# Bending Analysis of Multiferroic Semiconductor Composite Beam towards Smart Cement-Based Materials

**DOI:** 10.3390/ma16010421

**Published:** 2023-01-02

**Authors:** Yun Wang, Yifan Huang, Chunli Zhang, Rongqiao Xu

**Affiliations:** 1School of Mechanical Engineering, Hangzhou Dianzi University, Hangzhou 310018, China; 2Department of Engineering Mechanics, Zhejiang University, Yuquan Campus, Hangzhou 310027, China; 3Department of Civil Engineering, Zhejiang University, Zijingang Campus, Hangzhou 310058, China

**Keywords:** piezoelectric semiconductor, composite beam, multi-field coupling response, magnetoelectric effect, flexural deformation

## Abstract

A beam-like structure of antisymmetric laminated multiferroic piezoelectric semiconductor (LMPS), which consists of two piezomagnetic (PM) and two piezoelectric semiconductor (PS) layers is proposed. The structure could be in pure flexure deformation under an applied magnetic field. Through this deformation mode and the induced polarization field through the magneto-electro-semiconductive (MES) coupling mechanism, the semiconducting properties of PS layers can be manipulated by the applied magnetic field. In order to better understand and quantitatively describe this deformation mode, the one-dimensional governing equations for the LMPS beam are developed based on the three-dimensional theory. The analytical solutions are then presented for the LMPS cantilever beam with open-circuit conditions. The multi-field coupling responses of the LMPS cantilever beam under the longitudinal magnetic field are investigated. Numerical results show that the amplitude of each physical quantity is proportional to the applied magnetic field, and the thickness ratio of the PS phase plays a significant role in the MES coupling behaviors of the LMPS beam. The proposed structure can be integrated into cement structures but also fabricated cement-based multiferroic PS composite materials and structures. It provides an important material and structure basis for developing structural health monitoring systems in the fields of civil and transportation infrastructures.

## 1. Introduction

With the increasing demand for intelligent civil and transportation infrastructures, there is more and more research into cement-based electronic materials [1,2], including cement-based piezoelectric materials [3], cement-based piezoresistive materials [4], cement-based PN-junction [5], etc. These cement-based electronic materials can be used to realize various self-sensing and self-healing concrete components and structures [6], and thus have a huge potential application in the construction of intelligent civil and transportation infrastructures. Piezoelectric semiconductor (PS) materials are promising building blocks for multi-functional electronic devices due to the combination of piezoelectricity and semiconducting properties. For a PN or Schottky junction made of PS materials, the transport behavior of charges at the interface can be changed by a mechanical force via piezoelectricity; such a phenomenon is called a piezotronic effect [7,8,9,10]. The piezotronic effect has been utilized to improve the performances of devices including piezoelectric transistors [11,12], photodetectors [13], solar cells [14], and energy collectors [15,16].

It is fundamental to the development of, or improvement of, PS devices that conduct theoretical analysis of multi-field coupling behaviors of PS structures. From the viewpoint of continuum, such a problem belongs to the multi-field coupling mechanics of deformation, polarization, and carrier. There have been a lot of useful theoretical studies on PS structures, for example, the static extensional and flexural deformations of ZnO rods and beams under different conditions as well as free vibrations [17,18,19,20,21,22]. The extensional forced vibrations of ZnO rods have been investigated [23,24]. Liang et al. [25] presented the analytical expressions of critical loadings of a static buckling ZnO rod for different boundary conditions. Using the local mechanical and thermal loads, some researchers theoretically achieved the formation of a local barrier and well regions in a pure PS fiber [26,27]. In addition, by making use of the product properties of composites, Cheng et al. [28,29] proposed a sandwich-type PS composite structure made of pure piezoelectric dielectrics and silicon semiconductors and realized the manipulation of electrons in the silicon layer with a mechanical force.

Due to the magnetic field having good penetrability, some researchers have experimentally demonstrated the manipulation of light emissions and luminescence of optoelectronic devices made of PS materials by applying a magnetic field [30,31,32]. However, there is quite limited theoretical investigation into the tuning of multi-field coupling behaviors of PS structures with a magnetic field. Recently, for multiferroic composite semiconductor (MCS) rods made of piezomagnetic (PM), piezoelectric, and semiconductor materials, Cheng et al. [33] and Kong et al. [34] analyzed their piezotronic responses under the applied constant and time-dependent magnetic fields; such a phenomenon of controlling semiconductor properties with a magnetic field is named as the magneto-electro-semiconductor (MES) coupling effect. Subsequently, the tuning of the axial and local transverse magnetic fields on the electrical responses of the sandwiched PM/PS/PM rods through the MES effects was studied [35,36]. Yang et al. [37] investigated the coupled bending and extension of a PM/PS bilayer under a transverse magnetic field. Further, Liang et al. [38] studied the tuning effect of the magnetic field on the behavior of a piezoelectric PN junction between two PM layers via the MES effect.

This paper proposes a laminated multiferroic PS (LMPS) beam-like structure consisting of two outer PM and two inner PS layers. The four layers are identical in thickness. The two PS layers have opposite polarized directions, and the two PM layers have opposite magnetized directions. Such a structure is asymmetrical, and thus can undergo a pure flexure deformation without any other deformation coupled together, and consequently, has a great practical value in the field of devices. Based on the linearized three-dimensional equations of MCS materials, the one-dimensional governing equations for a LMPS beam are established. For the bending of the LMPS cantilever beam with open-circuit conditions, the corresponding analytical solutions are presented. The multi-field coupling responses of the LMPS cantilever beam under the longitudinal magnetic field through the MES effects are investigated in detail.

The rest of this paper is arranged as follows. Section 2 summaries the basic equations of n-type multiferroic composite semiconductors in three-dimensions. Section 3 formulates one-dimensional equations and corresponding analytical solutions for a LMPS cantilever beam. Section 4 presents the numerical results and discussion. Finally, the conclusions and prospects are given in Section 5.

## 2. Basic Equations of n-Type Multiferroic Composite Semiconductors

This section summarizes the three-dimensional (3D) basic equations of n-type MCS materials and structures. The equilibrium equations of motion, Maxwell’s equations for electric displacement and magnetic induction, and continuity equation of electrons, read
(1a)$$T_{ji,j}=\rho {\ddot{u}}_{i},$$
(1b)$$D_{i,i}=q(N_{D}^{+}-n),$$
(1c)$$B_{i,i}=0,$$
(1d)$$J_{i,i}^{n}=q\dot{n},$$
where $T_{ij}$ is the stress, $D_{i}$ the electric displacement, $B_{i}$ the magnetic induction, $J_{i}^{n}$ the electron current density, $u_{i}$ the mechanical displacement, ρ the material density, q the elementary charge, n the electron concentration, and $N_{D}^{+}$ the concentration of impurities of donors. The constitutive relations for the stress, electric displacement, magnetic induction, and current density are
(2a)$$T_{ij}=c_{ijkl}S_{kl}-e_{kij}E_{k}-h_{kij}H_{k},$$
(2b)$$D_{i}=e_{ikl}S_{kl}+\varepsilon_{ik}^{}E_{k}+\alpha_{ik}H_{k},$$
(2c)$$B_{i}=h_{ikl}S_{kl}+\alpha_{ik}E_{k}+\kappa_{ik}H_{k},$$
(2d)$$J_{i}^{n}=qn\mu_{ij}^{n}E_{j}+qD_{ij}^{n}n_{,j},$$
where $S_{ij}$ is the strain, $E_{i}$ the electric field, $H_{i}$ the magnetic field, $c_{ijkl}$ the elastic constant, $e_{kij}$ the piezoelectric constant, $\varepsilon_{ik}^{}$ the dielectric constant, $h_{kij}$ the piezomagnetic constant, $\kappa_{ik}$ the magnetic permeability, $\mu_{ij}^{n}$ the mobility, and $D_{ij}^{n}$ the diffusion constant. In addition, $\mu_{ij}^{n}$ and $D_{ij}^{n}$ satisfy the Einstein relation
(3)$$\frac{\mu_{ij}^{n}}{D_{ij}^{n}}=\frac{q}{k_{B}T},$$
where $k_{B}$ is the Boltzmann constant and T denotes the temperature in Kelvin. It should be mentioned that although the constitutive relations for stress of Equation (2a), electric displacement of Equation (2b), and magnetic induction of Equation (2c) are for multiferroic materials, it can be reduced to those for piezoelectric, piezomagnetic, and elastic materials through deleting the associated terms. For instance, the magnetic or electric field-related term should be deleted for piezoelectric or piezomagnetic materials. The strain and electric field can be written in the form of
(4a)$$S_{ij}=(u_{i,j}+u_{j,i})/2,$$
(4b)$$E_{i}=-\varphi_{,i}.$$
where φ denotes electric potential.

The first term in Equation (2c), representing the drift current, is nonlinear and thus leads to a challenge of theoretical analysis. Adopting the same linearized method as in ref [19], the electron concentration becomes
(5)$$n=n_{0}+\Delta n,$$
where $n_{0}=N_{D}^{+}$ denotes the initial concentration of electrons and Δn represents the incremental concentration of electrons. For the sake of simplicity, consider a uniformly doped semiconductor with a small perturbation of electrons, namely, $N_{D}^{+}$ is constant and $\Delta n<<n_{0}$. In this case, the current equation of Equation (2c) can be linearized as
(6)$$J_{i}^{n}=qn_{0}\mu_{ij}^{n}E_{j}+qD_{ij}^{n}{\left(\Delta n\right)}_{,j},$$
and Equations (1c) and (1d) become
(7a)$$D_{i,i}=-q\Delta n,$$
(7b)$$J_{i,i}^{n}=q\Delta \dot{n}.$$

## 3. Formulation of an LMPS Cantilever Beam

Consider an LMPS cantilever beam with length *L*, thickness 2*h*_2_, and width, as shown in Figure 1. Assume L≫b and $L\gg 2h_{2}$, namely, the considered beam is slender. The geometric middle plane of the LMPS beam is chosen as the *o*-*x*_2_*x*_3_ coordinate plane. The LMPS beam is fixed at the left end. The two ends of the LMPS beam are with open-circuit conditions. Under the longitudinal constant magnetic field of $H_{3}$, the top and bottom PM layers undergo the tensile and compression deformations, respectively, along the *x*_3_ direction through the PM coupling effect. Subsequently, such a deformation is transferred to the PS layers through the two interfaces; thus, the piezoelectric polarization field appears in the PS layers due to piezoelectricity and leads to the redistribution of electrons as a result.

For the bending of the LMPS cantilever beam, the deflection $u_{1}$, axial displacement $u_{3}$, electric potential φ, and incremental concentration of electrons Δn can be approximated as [39]
(8a)$$u_{1}=v(x_{3},t),\;u_{3}=x_{1}\psi (x_{3},t),$$
(8b)$$\varphi =\varphi (x_{3},t),\;\Delta n=\Delta n(x_{3},t).$$
where v and ψ denote the deflection and rotation of the beam, respectively. From Equations (4a), (4b), (8a) and (8b), the non-zero strain and electric field components can be given as
(9a)$$S_{3}=S_{33}=u_{3,3}=x_{1}\psi ,$$
(9b)$$S_{5}=2S_{13}=u_{1,3}+u_{3,1}=v_{,3}+\psi ,$$
(9c)$$E_{3}=-\varphi_{,3}.$$

For the classical flexure, the shear deformation can be negligible, that is,
(10)$$S_{5}=v_{,3}+\psi ≅0.$$

The stress and electric displacement in the upper and lower PM layers keep the same form except for the effective piezomagnetic constant with opposite signs and are given by
(11a)$$T_{3}^{}={\bar{c}}_{33}^{PM}S_{3}\mp {\bar{h}}_{33}^{PM}H_{3},$$
(11b)$$D_{3}^{}={\bar{\varepsilon }}_{33}^{PM}E_{3},$$
where the effective material constants are
(12a)$${\bar{c}}_{33}^{PM}=c_{33}^{}-\frac{2c_{13}^{}c_{13}^{}}{c_{11}^{}+c_{13}^{}},$$
(12b)$${\bar{h}}_{33}^{PM}=h_{33}^{}-\frac{2c_{13}^{}h_{31}^{}}{c_{11}^{\left(2\right)}+c_{12}^{\left(2\right)}},$$
(12c)$${\bar{\varepsilon }}_{33}^{PM}=\varepsilon_{33}^{}.$$

Similarly, the stress and electric displacement in the upper and lower PS layers have the same form except for the effective piezoelectric constant with opposite signs and read
(13a)$$T_{3}^{}={\bar{c}}_{33}^{PS}S_{3}\mp {\bar{e}}_{33}^{PS}E_{3},$$
(13b)$$D_{3}^{}=\pm {\bar{e}}_{33}^{PS}S_{3}+{\bar{\varepsilon }}_{33}^{PS}E_{3},$$
where the effective material constants are
(14a)$${\bar{c}}_{33}^{PS}=c_{33}^{}-\frac{2c_{13}^{}c_{13}^{}}{c_{11}^{}+c_{12}^{}},$$
(14b)$${\bar{e}}_{33}^{PS}=e_{33}^{}-\frac{2c_{13}^{}e_{31}^{}}{c_{11}^{}+c_{12}^{}},$$
(14c)$${\bar{\varepsilon }}_{33}^{PS}=\varepsilon_{33}^{}+\frac{2e_{31}^{}e_{31}^{}}{c_{11}^{}+c_{12}^{}}.$$

The electron current density in the PS layer is
(15)$$J_{3}^{n}=-qn_{0}\mu_{33}^{n}\varphi_{,3}+qD_{33}^{n}{\left(\Delta n\right)}_{,3}.$$

Following the same routine in ref [33], the governing equations
(16a)$$M_{,33}=0,$$
(16b)$$D_{,3}=-2bh_{1}q\Delta n,$$
(16c)$$J_{3,3}^{n}=0.$$
are obtained for LMPS beams and where *M* and *D* denote the bending moment and resultant electric displacement, respectively. They are defined by
(17a)$$M=\int_{A}x_{1}T_{3}dA=\hat{c}\psi_{,3}+\hat{e}\varphi_{,3}-\hat{h}H_{3},$$
(17b)$$D=\int_{A}D_{3}dA=-\hat{\varepsilon }\varphi_{,3}+\hat{e}\psi_{,3},$$
where
(18a)$$\hat{c}=\frac{2b(h_{2}^{3}-h_{1}^{3})}{3}{\bar{c}}_{33}^{PM}+\frac{2bh_{1}^{3}}{3}{\bar{c}}_{33}^{pS},$$
(18b)$$\hat{e}=bh_{1}^{2}{\bar{e}}_{33}^{PS},\;\hat{h}=b(h_{2}^{2}-h_{1}^{2}){\bar{h}}_{33}^{PM},$$
(18c)$$\hat{\varepsilon }=2b(h_{2}-h_{1}){\bar{\varepsilon }}_{33}^{PS}-2bh_{1}{\bar{\varepsilon }}_{33}^{PM}.$$

Substitution of Equations (15), (17a) and (17b) into Equations (16a)–(16c) yields
(19a)$$\hat{c}\psi_{,333}+\hat{e}\varphi_{,333}=0,$$
(19b)$$-qn_{0}\mu_{33}^{n}\varphi_{,33}+qD_{33}^{n}\Delta n_{,33}=0,$$
(19c)$$-\hat{\varepsilon }\varphi_{,33}+\hat{e}\psi_{,33}=-2bh_{1}q\Delta n.$$

Eliminating the rotation ψ and incremental concentration of electrons Δn from Equations (19a)–(19c), the equation satisfied by the electric potential is obtained as
(20)$$\varphi_{,3333}-a_{1}\varphi_{,33}=0,$$
where
(21)$$a_{1}=\frac{2bh_{1}\hat{c}}{\hat{\varepsilon }\hat{c}+\hat{e}\hat{e}}\frac{qn_{0}\mu_{33}^{n}}{D_{33}^{n}}.$$

The electric potential in Equation (20) has a general solution in the form of
(22)$$\varphi =C_{1}\sinh\lambda_{1}x_{3}+C_{2}\cosh\lambda_{1}x_{3}+C_{3},$$
where $C_{1}$, $C_{2}$, and $C_{3}$ are undetermined constants. Substituting Equation (22) into Equations (19a) and (19c) gives the analytical solutions of ψ and Δn as
(23a)$$\psi =-\frac{\hat{e}}{\hat{c}}\left(C_{1}\sinh\lambda_{1}x_{3}+C_{2}\cosh\lambda_{1}x_{3}\right)+C_{4}x_{3}+C_{5}-\frac{\hat{e}}{\hat{c}}C_{3},$$
(23b)$$\Delta n=\frac{\hat{\varepsilon }\hat{c}+{\hat{e}}^{2}}{2bh_{1}\hat{c}q}\lambda_{1}^{2}\left(C_{1}\sinh\lambda_{1}x_{3}+C_{2}\cosh\lambda_{1}x_{3}\right),$$
where $C_{4}$ and $C_{5}$ are unknown constants. Then, by substituting ψ of Equation (23a) into Equation (10), the deflection of the LMPS beam is obtained as
(24)$$v=\frac{\hat{e}}{\lambda_{1}\hat{c}}\left(C_{1}\cosh\lambda_{1}x_{3}+C_{2}\sinh\lambda_{1}x_{3}\right)-\frac{C_{4}}{2}x_{3}^{2}-C_{5}x_{3}+\frac{\hat{e}}{\hat{c}}C_{3}x_{3}+C_{6},$$
where *C*_6_ is an unknown constant.

The six undetermined constants $C_{i}$ are determined by the fixed-free boundary conditions and open-circuit conditions at the two ends as well as the choice of the zero point for the electric potential, namely,
(25a)ψ(0)=0,     v(0)=0,     φ(0)=0,
(25b)M(L)=0,     D(0)=0,     D(L)=0.

## 4. Numerical Results and Discussion

It is supposed that the two inner PS and outer PM layers of the LMPS beam are made of n-type ZnO and CoFe_2_O_4_, respectively, whose material properties are listed in Table 1. The initial electron concentration of ZnO is $n_{0}=1\times 10^{21}$ m^−3^. For the considered beam, the length is *L* = 3 μm, the total thickness is 2*h*_2_ = 0.2 μm, and the width is *b* = 0.2 μm. Denote $h_{1}/h_{2}$ as δ, which is the thickness ratio of the PS phase to the total thickness of the beam. With the analytical solutions derived above, this example evaluates the macroscopic multi-field coupling responses of the LMPS beam under different applied magnetic fields.

First, consider the cases of the LMPS beam with δ=0.5 under three different magnetic fields (*H*_3_ = 500 A/m, 1000 A/m, and 1500 A/m). Figure 2 shows the distributions of the deflection, electric potential, electric displacement, and incremental concentration of electrons along the beam. It can be observed from Figure 2a that a strong magnetic field leads to a large deflection or deformation as expected, and then the larger electric potential, electric displacement, and incremental concentration of electrons are induced in the beam though the piezoelectric and MES coupling effects. Figure 2b,d show that the distribution of Δn keeps the similar trend as that for the electric potential. Δn changes sharply near the two ends of the beam and achieves its negative and positive peak values, respectively, at the left and right ends of the beam, while approaching zero in the middle part of the beam.

Figure 3 shows the curves of the deflection v(L), electric potential φ(L), incremental concentration of electrons Δn(L), and $\Delta n(L)/n_{0}$ at the right end of the beam versus the applied magnetic field. It can be seen that the amplitudes of v(L), φ(L), and Δn(L) are directly proportional to the applied magnetic field. This indicates that the performance of the proposed LMPS structure could be effectively manipulated by an applied magnetic field. At the same time, the proposed LMPS structure can serve as a sensor for monitoring the change of the surrounding magnetic field. Figure 3d, shows that the incremental concentration of electrons is smaller than 3% of the initial concentration of electrons, which is satisfied with the assumption of linearization; hence, the numerical results obtained by the linearized model are accurate and reliable for the considered problem in this paper.

Next, by fixing the total thickness 2*h*_2_ = 0.2 μm and the initial concentration of electrons *n*_0_ = 10^21^ m^−3^, we investigate the multi-field coupling responses of the LMPS beam with four different thickness ratios of δ=0.3, δ=0.5, δ=0.7, and δ=0.9 under the applied magnetic field of *H*_3_ = 1000 A/m. The distributions of the deflection, electric potential, electric displacement, and incremental concentration of electrons along the beam are plotted in Figure 4, from which it can be seen that the thickness ratio of the PS phase has a considerable effect on the macroscopic multi-field coupling responses of the LMPS beam. The electric potential φ(L) at $x_{3}=L$ for four different δ increases as in the sequence of δ=0.7, δ=0.5, δ=0.9, and δ=0.3. In order to obtain the optimum thickness ratio of δ for the proposed LMPS structure, φ(L) and Δn(L), $\alpha_{E}$, and $\alpha_{n}$ with δ ranging from 0 to 1 are evaluated and plotted in Figure 5. It is observed that when δ is about 0.72, both the ME and MES coupling effects reach their peaks. In this example, the optimum region of δ is defined such that the value of $\alpha_{E}$ or $\alpha_{n}$ is not less than about 90% of the corresponding peak value. In this sense, Figure 5 shows that the optimum region of δ ranges from about 0.6 to 0.8 for the proposed LMPS beam.

## 5. Conclusions and Prospects

This paper proposes an antisymmetric LMPS composite beam consisting of two opposite magnetized PM layers and two opposite polarized n-type PS layers. Under the longitudinal magnetic field, the LMPS cantilever beam undergoes a pure flexure deformation. The one-dimensional equations for the LMPS cantilever beam as well as its analytical solutions for open-circuit conditions are presented. According to this analytical solution, the multi-field coupling or piezotronics responses of the LMPS beam can be manipulated by the longitudinal magnetic field through the MES coupling effect. As the linearized model is employed, the evaluated response of each physical quantity is proportional to the applied magnetic field. The results show that the thickness ratio of the PS phase has a significant effect on its multi-field coupling behaviors, and the optimum thickness ratio of the PS phase locates in the range of 0.6–0.8 or so. It means the proposed LMPS structure is optimal for magnetic field-related smart devices, which can be used to remotely manipulate the performance of engineering structures as well as monitor its surrounding magnetic fields without contact. Hence, the proposed LMPS can be integrated into cement structures and also fabricated cement-based multiferroic PS composite materials and structures through special process. It brings a bright application prospect for the development of non-contact, self-sensing, and active-control technology in a structure’s state, such as stress and deformation, for the cement-based structures of civil and transportation infrastructures. In addition, for such a laminated structure, the interfaces could not be bonded together perfectly, which will have a significant effect on the multi-field coupling mechanical responses of LMPS structures, and a theoretical model incorporating an imperfect interface should be developed. It is highly desirable to fabricate such cement-based LMPS composite structures and conduct experimental studies based on the theoretical results obtained in this paper. However, there are no relevant experimental conditions in our group; we are hoping for such a cooperation with other researchers in the future.

## Figures and Tables

**Figure 1 materials-16-00421-f001:**
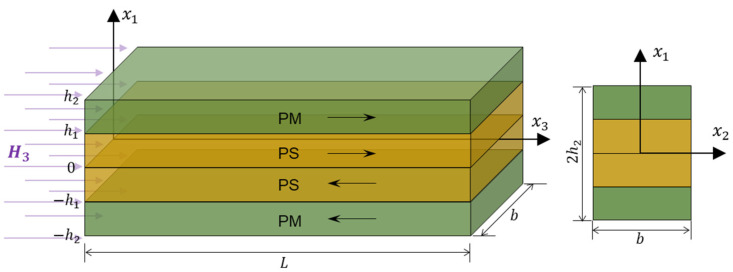
Sketch of an LMPS beam under the longitudinal magnetic field.

**Figure 2 materials-16-00421-f002:**
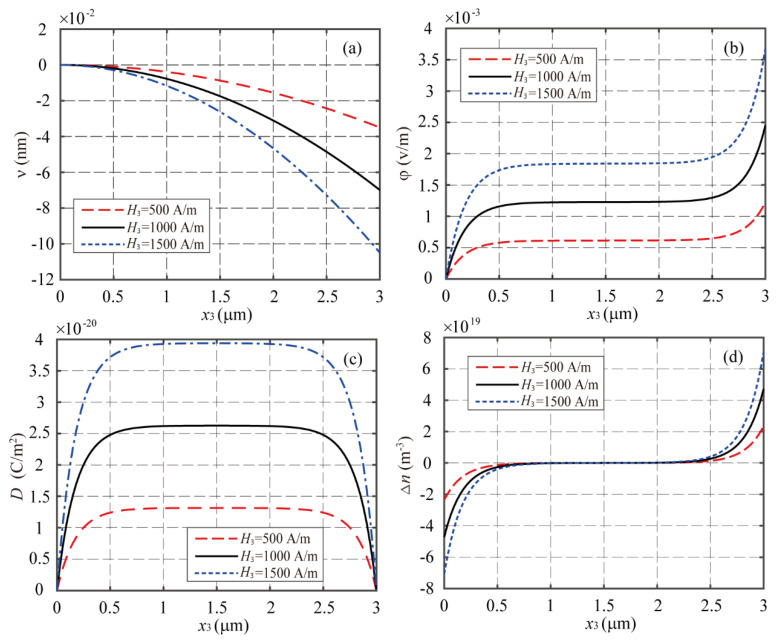
The distributions of (**a**) *v*, (**b**) φ, (**c**) *D*, and (**d**) Δ*n* along the LMPS cantilever beam with n_0_ = 10^21^ m^−3^ for different magnetic fields.

**Figure 3 materials-16-00421-f003:**
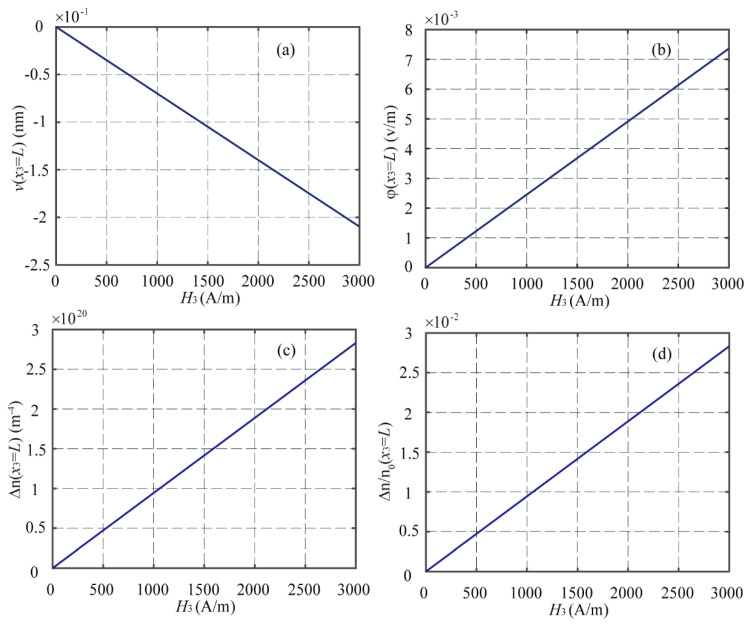
The values of (**a**) *v*, (**b**) *φ*, (**c**) Δn, and (**d**) $\Delta n/n_{0}$ at *x*_3_ = *L* of the LMPS beam with *n*_0_ = 10^21^ m^−3^ under different magnetic fields.

**Figure 4 materials-16-00421-f004:**
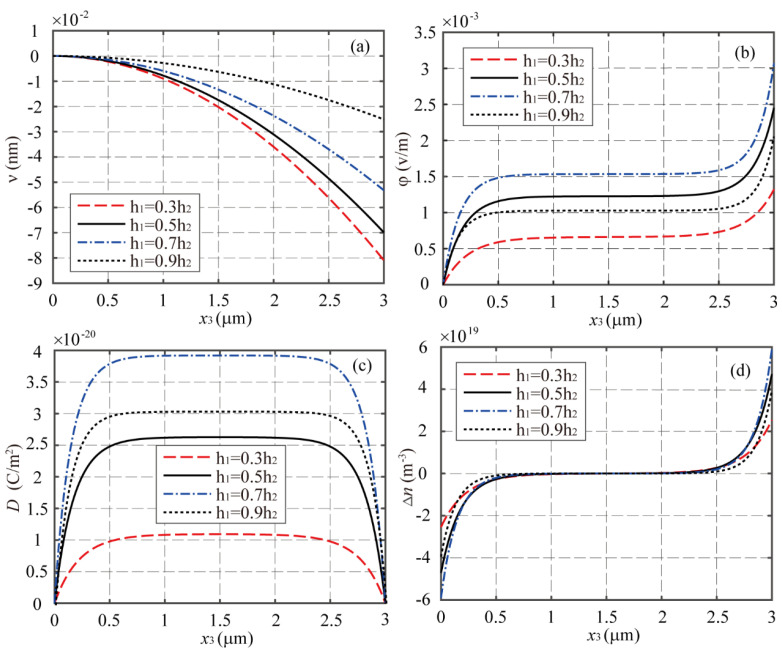
The distributions of (**a**) *v*, (**b**) φ, (**c**) *D*, and (**d**) Δ*n* along the LMPS beam with *n*_0_ = 10^21^ m^−3^ and *H*_3_ = 1000 A/m for different δ.

**Figure 5 materials-16-00421-f005:**
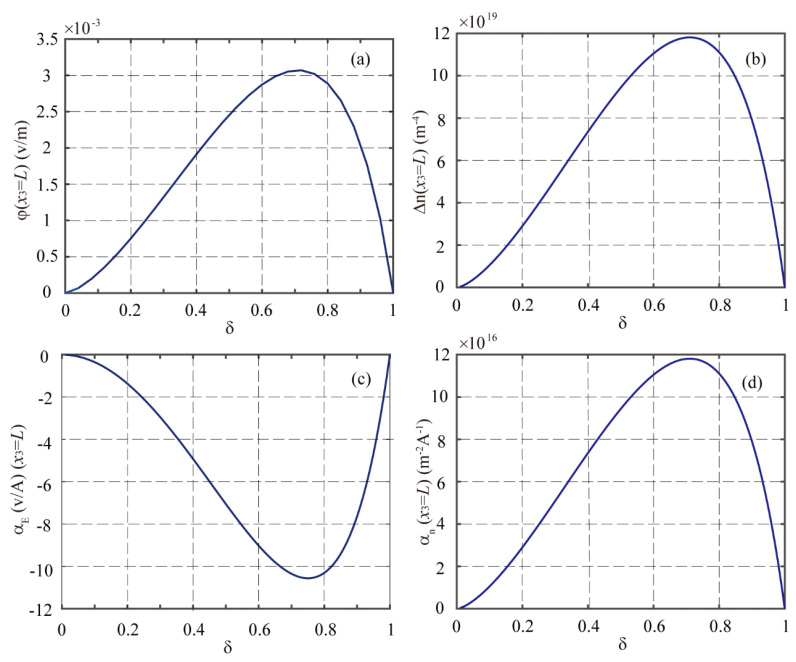
The values of (**a**) φ, (**b**) Δn, (**c**) $\alpha_{E}$, and (**d**) $\alpha_{n}$ at *x*_3_ = *L* of the LMPS beam with *n*_0_ = 10^21^ m^−3^ and *H*_3_ = 1000 A/m for different δ.

**Table 1 materials-16-00421-t001:** Material constants of ZnO and CoFe_2_O_4._

	ZnO [40]	CoFe_2_O_4_ [41]
Elastic constant (GPa)	$c_{33}^{}=1.45$, $c_{44}^{}=42.4$	$c_{33}^{}=269$, $c_{44}^{}=45.3$
Piezoelectric constant (C/m^2^)	$e_{33}^{}=1.691$, $e_{15}^{}=-0.481$	-
Piezomagnetic constant (N/A·m)	-	$h_{33}^{}=799.7$
Dielectric constant (10^−11^ C/N·m^2^)	$\varepsilon_{22}^{}=8.11$, $\varepsilon_{33}^{}=11.2$	$\varepsilon_{22}^{}=8$, $\varepsilon_{33}^{}=9.8$

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
