# Peer review of "Bending Analysis of Multiferroic Semiconductor Composite Beam towards Smart Cement-Based Materials"

_materials, 2023, doi:10.3390/ma16010421_

Round 1
Reviewer 1 Report
This study aims to do an investigation on the bending performance of anti-symmetric multiferroic piezoelectric semiconductor composite beams towards smart cement-based materials. f packing density and water film thickness in the synergistic effects of slag and silica fume. The paper is interesting and could be accepted for publication after a major revision. So, the authors are invited to address the following comments carefully.
1- The title is too long and it is not straightforward. So, please find a more proper title for your paper.
2- The abstract need to be improved significantly. The authors should discuss their study in more detail. Also, the most important results should be presented at the end of the abstract
3- A list of notions is recommended to be provided.
4- The paper needs to polish in terms of English.
5- The introduction is so short. So, more studies should be discussed in detail to show the gap in previous studies that resulted in doing current investigation
6- This is strongly recommended to provide a “research significance” section to talk about the novelty of this study in detail
7- Please divide your equations. For example, equation 2 should be divided into four
8- Also, the explanation of the equations is not clear and difficult to understand for a reader
9- This is recommended to replace Figure 1 with a high-quality 3D figure
10- More benchmarks under a combination effect of various loads should be utilized in Section 4. Otherwise, the presented results are not sufficient to consider this paper for a high-quality journal
11- The error of presented modelling with the existing formulations should be presented to highlight the accuracy of the current study in comparison with other previous research.
12- Also, the presented results should be compared with exact answers which could be utilized by FEM.
13- The conclusion section should be extensively rewritten and both quantity and quality analysis should be presented and discussed.
So, the paper is recommended for publication after addressing the comments carefully with a major revision.
Reviewer 2 Report
In the present paper, a cement-based anti-symmetric multiferroic piezoelectric semiconductor composite beam has been proposed. A theoretical investigation has been carried out to get the new results. The paper needs a revision before reconsideration. The reviewer comments are suggested as follows:
1. Authors may explain deficiencies or shortcomings of other studies to make a bridge to introducing the novelty of their work. The novelty of this work must be more explained.
2. The application of the proposed model needs to be discussed in detail.
3. Abstract and concluding remarks need to be revised as it has been written in a very ambiguous manner.
4. The introduction should end with an overview/outline of the rest of the paper.
5. Throughout the manuscript, several grammatical mistakes are evident. Some sentences are also not clear. The authors need to revise it thoroughly.
6. Why flexural analysis is required in this case, and how it influences the overall performance? Please explain.
7. More physical descriptions should be added to the results and discussion section.
8. No validation study has been presented. Authors need to present a few validation studies to ascertain the authenticity and efficacy of the developed mode.
9. The developed model consists of two piezomagnetic (PM) and two piezoelectric semiconductor (PS) layers. Both layers have different properties. How does the developed model accommodate this disparity of properties at the interface?
10. Proper citations of various equations need to be given at the appropriate places.
Round 2
Reviewer 2 Report
Accepted